# Performance Assessment of Software to Detect and Assist Prescribers with Antimicrobial Drug Interactions: Are all of them Created Equal?

**DOI:** 10.3390/antibiotics9010019

**Published:** 2020-01-04

**Authors:** Elena Morte-Romea, Pilar Luque-Gómez, Mercedes Arenere-Mendoza, Jose Luis Sierra-Monzón, Ana Camón Pueyo, Galadriel Pellejero Sagastizabal, Guillermo Verdejo Muñoz, David Sánchez Fabra, José Ramón Paño-Pardo

**Affiliations:** 1Infectious Diseases Service, Hospital Clínico Universitario “Lozano Blesa”, 50009 Zaragoza, Spain; emromea@gmail.com; 2Instituto de Investigación Sanitaria Aragón (IIS Aragón), 50009 Zaragoza, Spain; pluque@salud.aragon.es (P.L.-G.); marenere@salud.aragon.es (M.A.-M.); jlsierram@gmail.com (J.L.S.-M.); anacamon0211@gmail.com (A.C.P.); alamismami@hotmail.com (G.P.S.); guillemovil@hotmail.com (G.V.M.); davidsanchezfabra@gmail.com (D.S.F.); 3Intensive Care Unit. Hospital Clínico Universitario “Lozano Blesa”, 50009 Zaragoza, Spain; 4Department of Hospital Pharmacy, Hospital Clínico Universitario “Lozano Blesa”, 50009 Zaragoza, Spain; 5Emergency Room Department, Hospital Clínico Universitario “Lozano Blesa”, 50009 Zaragoza, Spain; 6Internal Medicine Department, Hospital Clínico Universitario “Lozano Blesa”, 50009 Zaragoza, Spain

**Keywords:** antimicrobial, antimicrobial stewardship, antimicrobial drug interactions, pharmacy

## Abstract

Background: Detecting and managing antimicrobial drug interactions (ADIs) is one of the facets of prudent antimicrobial prescribing. Our aim is to compare the capability of several electronic drug–drug interaction (DDI) checkers to detect and report ADIs. Methods: Six electronic DDI checking platforms were evaluated: Drugs.com^®^, Medscape^®^, Epocrates^®^, Medimecum^®^, iDoctus^®^, and Guía IF^®^. Lexicomp^®^ Drug Interactions was selected as the gold standard. Ten ADIs addressing different mechanisms were evaluated with every electronic DDI checker. For each ADI, we assessed five dimensions and calculated an overall performance score (maximum possible score: 10 points). The explored dimensions were sensitivity (capability to detect ADI), clinical effect (type and severity), mechanism of interaction, recommended action(s), and documentation (quality of evidence and availability of references). Results: The electronic DDI checkers did not detect a significant proportion of the ADI assessed. The overall performance score ranged between 4.4 (Medimecum) and 8.8 (Drugs.com). Drugs.com was the highest ranked platform in four out of five dimensions (sensitivity, effect, mechanism, and recommended action). Conclusions: There is significant variability in the performance of the available platforms in detecting and assessing ADI. Although some ADI checkers have proven to be very accurate, others missed almost half of the explored interactions.

## 1. Introduction

The ageing of the population and the increasing burden of chronic diseases are some of the major drivers that have led to incremental polypharmacy in the last few years. Polypharmacy can increase the risk of adverse drug events (ADEs), becoming an issue that can prolong hospitalization and increase hospital admissions and emergency room visits. It is estimated that drug–drug interactions (DDIs) are responsible for 3 to 5% of preventable ADEs [1].

Antimicrobials can be involved in multiple DDIs. These are called antimicrobial drug interactions (ADIs) and may produce undesirable harm to the patient, either by decreasing antibiotic efficacy or by favoring toxic side effects. As antimicrobials are among the most frequently prescribed drugs, detecting and minimizing the impact of ADIs is important. Moreover, with the rise of multidrug-resistant microorganisms and the need to use sometimes less efficacious and more toxic second line drugs, preventing ADI is even more relevant.

The amount and complexity of DDIs are of such magnitude that physicians and pharmacists need assistance to optimize DDI detection and subsequent management; for example, Glassman et al. found that clinicians could only detect 44% of the possible DDI pairs (range 11–64%) [2].

Several computerized DDI tools that detect and rank the severity of DDIs are available, offering therapeutic alternatives. While these tools are highly desirable, their value depends on how sensitive they are in detecting DDIs and on their accuracy in assessing the type and severity of the interactions and, thus, need to be validated. Some of these tools have been evaluated in different studies, finding conflicting results. For example, Hazlet et al. found that up to 33% of the preventable and relevant DDIs were not recognized by these platforms [3].

Although electronic platforms and tools for DDI assessment have been validated in several studies, we have not found any studies focusing specifically on ADIs [4,5]. Our aim is to compare the performance of several drug–drug interaction software platforms to detect and characterize ADIs.

## 2. Material and Methods

We designed a strategy for the performance assessment of several DDI software platforms using real-life examples of ADIs.

### 2.1. Drug–Drug Interaction Software Platforms Evaluated

The performance of six software platforms for DDI assessment was evaluated. The DDI software platforms were selected by the research team based on their popularity among the antimicrobial stewardship team members as well as among other local physicians and pharmacists. Nevertheless, to be eligible, the DDI software platforms had to: (1) Be either accessible online from a desktop computer or through a smartphone app; (2) be available in English or Spanish; and (3) allow multiple simultaneous DDI checking. The DDI software platforms included were: Drugs.com^®^ v2.8.2 professional option 2, Medscape^®^ v5.10, Epocrates online^®^, Medimecum^®^ v1.25, iDoctus^®^ v2.2.401, and Guía IF^®^ v1.0.4.2. The latest versions of each software platform were downloaded or consulted from the publisher´s website. Lexicomp Drug Interactions^®^ v3.18.0 was selected as the gold standard, since it has been shown to have appropriate sensitivity and specificity for detecting DDIs elsewhere [6], and also having been evaluated in our study.

### 2.2. Selection of Antimicrobial-Drug Interactions to be Evaluated

Ten ADIs were chosen to be checked by each of the DDI software platforms. The ADIs were selected by the researchers EMR and JRP from a list of potential ADIs obtained from a reference pharmacology textbook [7]. All ADIs were considered to be clinically relevant, involved frequently prescribed drugs and covered the main mechanisms of DDI (See Table 1).

### 2.3. Performance Assessment

Each ADI was checked in all of the DDI software platforms by six healthcare professionals that usually deal with ADIs in their daily practice (a hospital pharmacist, an ER physician, an internal medicine and infectious diseases physician, and three internal medicine residents). 

The performance of the DDI software platforms to detect and assist prescribers with ADIs was assessed from three different perspectives, which were accuracy (ability to detect the ADI and describe the type of toxicity), completeness (information about severity, mechanism of interaction, clinical management, level of evidence and reference availability), and user experience (subjective impression of usability and clarity).

A performance score was designed considering the following dimensions: sensitivity, clinical effect (type of toxicity and severity), mechanism of interaction, recommended action(s), and documentation (quality of evidence and availability of references). The maximum possible score was 10 points. A checklist was developed to standardize the comparison with the gold standard. 

In order to evaluate the user experience, the perceived usability of the software platform (easiness to use) and the clarity (being intelligible) of the explanation of the mechanism of interactions were assessed. 

### 2.4. Data Analysis

Basic descriptive statistics were used to calculate the scores. The DDI platform characteristics were assigned a rating in proportion to their relevance, reaching a maximum overall score of 10 points. The sensitivity was defined as the capability of the software to detect the ten selected ADIs (expressed as a percentage), scoring from 0 points (the DDI checker detects less than 6 ADIs) to 3 points (the DDI checker detects all ADIs). The dimension recommended action and clinical effect were evaluated over 3 and 2 points respectively (if they are generated by the software and are similar to the ones offered by the gold standard). The other characteristics (mechanism, quality of evidence, and availability of references) only scored if they were generated by the evaluated software. The final score in each DDI software checker dimension was reached by adding up the partial score for each selected ADI over the detected ADIs by the platform, in proportion to the predefined maximum score for the dimension (See Appendix A). 

## 3. Results

All but one of the evaluated platforms were freely accessible. The researchers were granted unrestricted temporary access to Medimecum^®^, the only software platform accessible through paid subscription. Drugs.com and Epocrates were accessible only via the Internet on desktop computers, while Guía IF and Medimecum were available via a smartphone app, exclusively. Medscape and iDoctus were accessible via the Internet and a mobile app. The main capabilities of each DDI software platform are described in Table 2.

Sensitivity ranged from 40 (Guía IF) to 100% (Drugs.com), as displayed in Table 3. Drugs.com obtained the highest overall performance score (8.8/10) while the lowest score was obtained by Medimecum (4.4/10), as shown in Table 4. No DDI software platform reached the maximum possible score in the “clinical effect” and “recommended action(s)” dimensions. Drugs.com was the platform that reached the highest score in both categories. Neither Epocrates online nor Medimecum provided information concerning the severity of the ADIs. Most DDI software platforms provided information on the mechanism of interaction, although the information provided varied remarkably between them. Only Drugs.com, Lexicomp Drug Interactions, and iDoctus presented a reference list (Table 1).

The perceived usability of the DDI software platforms ranged from two (Guia IF) to four (Drugs.com and iDoctus). iDoctus and Drugs.com obtained the maximum possible score in regard to perceived clarity presenting the mechanism of interaction and the recommendations (Figure 1). 

## 4. Discussion

Antibiotic drug interactions pose a significant opportunity to improve efficacy and to decrease the adverse events associated with antimicrobial therapy, the main aims of antimicrobial stewardship programs [6]. Checking for interactions should be part of good prescribing habits, but given the unmanageable number of potential pairs of interactions, electronic tools, namely DDI software platforms, can be of great help to accomplish this task. To the best of our knowledge no antibiotic-focused systematic evaluation of electronic DDI checkers had been conducted to date.

In this study, we found remarkable differences in the performance of electronic DDI checkers with regard to the detection of ADIs and the assistance to healthcare professionals by providing recommendations on how to manage these types of interactions. One of the most relevant findings is the heterogenous capability of these tools to detect ADIs. Indeed, some of them failed to detect up to half of the tested ADIs. Similarly, significant differences were found among the DDI software platforms when assessing the clinical effect and formulating the recommended action(s), as compared with the gold standard. Interestingly, the user´s perceptions (usability and clarity) did not necessarily match the overall performance status of the software platforms. 

The capability to detect ADIs (sensitivity) is probably the most important performance measurement for DDI software platforms since it is a limiting step; if an ADI is not detected, no further action can be taken. Despite its paramount relevance, DDI software platforms have shown significant variations in overall sensitivity, ranging from 26 to 100% of all DDI evaluated [5,7,8,9]. As with sensitivity, other relevant performance dimensions have been found to vary significantly between electronic checkers. 

There were several limitations in this study that should be acknowledged. First, although the main mechanisms of ADIs have been considered, we did not perform an exhaustive evaluation, limiting the assessment to ten ADIs that were considered strategic. Second, although based on the previous literature and personal clinical judgment we used five dimensions to assess the electronic DDI checkers, the assessment of these tools has not been standardized yet. Also, the selection of the gold standard might be questionable, since several others could have been considered, too. Finally, although we used a classic approach to assess the DDI software platforms, we did not explore how these resources impact real-life decision-making.

We believe that this study points out several opportunities to improve. In the absence of the systematic validation of electronic DDIs checkers, user experience, mainly usability, may well be the ultimate factor influencing clinicians’ decisions when choosing between the multiple available resources. This, indeed, can lead to suboptimal therapeutic decisions since usability does not necessarily match with accuracy, and could be mitigated if these resources were validated first, before making it openly available. Regulatory bodies such as U.S. Food and Drug Administration (FDA) and European Medicine Agency (EMA) may play an essential role. Nevertheless, despite the FDA encouraging the development of mobile medical applications or computing platforms, we are not aware of initiatives that include the systematic evaluation and/or the validation of electronic DDI checkers [10]. Indeed, more active involvement of regulatory agencies in the evaluation of these resources could lead to a better and more homogeneous performance of these resources.

## 5. Conclusions

To conclude, we have found significant differences in the sensitivity and accuracy of six electronic DDI checkers when assessing ADIs, which are likely to lead to suboptimal prescribing decisions. The validation and public reporting of performance status of these resources could contribute to solving this problem. 

## Figures and Tables

**Figure 1 antibiotics-09-00019-f001:**
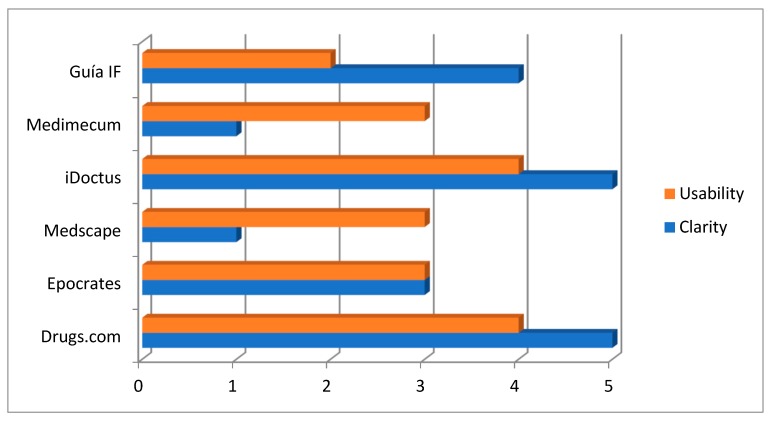
Perceived usability of the software platform and clarity of the mechanism of action and the recommendations (0: Awful; 1: Poor; 2: Okay; 3: Good; 4: Very good; 5: Excellent).

**Table 1 antibiotics-09-00019-t001:** Antimicrobial drug interactions selected to be tested.

Mechanism of interaction	Antimicrobial Drug Interactions
**Pharmacodynamic Interactions**	
Boosting of effects on electrolyte renal excretion	Angiotensin-Converting Enzyme Inhibitors/Trimethoprim
Increased net effect of neurotransmission	Metoclopramide/Serotonin Modulators (Linezolid) Quinolones/Nonsteroidal Anti-Inflammatory Agents (NSAIDs)
Enhancement of musculoskeletal adverse effects	Quinolones/Corticosteroids (Systemic)
Prolongation of the QT interval	Escitalopram/Levofloxacin
**Pharmacokinetic Interactions**	
Impaired absorption	Cefditoren/Proton Pump Inhibitors Quinolones/Antacids
Enhanced metabolism	Carbapenemes/Valproic Acid (unknown pathway) Rifampicin/Apixaban (induction of CYP3A4)
Impaired metabolism	Fluconazole/Calcium Channel Blockers (inhibition of CYP3A4)

**Table 2 antibiotics-09-00019-t002:** Capabilities and main features of the selected drug–drug interaction software platforms.

DDI Software Platform	Lexicomp Drug Interactions	Drugs.com	Epocrates Online	Medscape	iDoctus	Medimecum	Guía IF
**Language**	English	English	English	English	Spanish	Spanish	Spanish
**Online / Offline ***	Online	Online	Online	Online	Offline	Offline	Offline
**Clinical effect**	Yes	Yes	Yes	Yes	Yes	Sometimes	Sometimes
**Severity**	Yes	Yes	No	Sometimes	Yes	No	Sometimes
**Mechanism of interaction**	Yes	Yes	Yes	Yes	Yes	No	Sometimes
**Recommendations for clinical management**	Yes	Yes	Yes	Yes	Yes	Yes	Sometimes
**Display of the level of evidence**	Yes	No	No	No	Yes	No	No
**Reference list**	Yes	Yes	No	No	Yes	No	Sometimes
**Date of last update ****	2019	02/12/2019	Not available	11/12/2017	Not available	Not available	15/04/2014
**Source**	Wolters Kluwer Clinical Drug Information	Micromedex^®^ Cerner Multum Wolters Kluwer American Society of Health System Pharmacists	Athenahealth ^®^	Medscape Publishers’ Circle^®^	Consejo General de Colegios Oficiales Farmacéuticos	Springer Healthcare Ibérica^®^	Sociedad Española de Farmacia Hospitalaria

* Online means platform is only available with Internet connection (desktop computer or smartphone app). Offline means the platform can be used without Internet connection (desktop computer or smartphone app). ** The last update indicated in each platform, checked on the 21st of December 2019.

**Table 3 antibiotics-09-00019-t003:** Ability of drug–drug interactions (DDIs) software platforms to detect antimicrobial-drug interactions (ADIs).

DDI Software Platform	Lexicomp Drug Interactions	Drugs.com	Epocrates online	Medscape	iDoctus	Medimecum	Guía IF
Trimethoprim/Enalapril	Yes	Yes	Yes	Yes	Yes	**No**	**No**
Linezolid/Metoclopramide	Yes	Yes	Yes	**No**	**No**	**No**	**No**
Levofloxacin/Systemic prednisone	Yes	Yes	Yes	Yes	**No**	Yes	**No**
Escitalopram/Ciprofloxacin	Yes	Yes	**No**	**No**	Yes	Yes	**No**
Levofloxacin/antacids	Yes	Yes	Yes	**No**	Yes	Yes	Yes
Meropenem/valproic acid	Yes	Yes	Yes	Yes	Yes	Yes	**No**
Quinolones/NSAIDs	Yes	Yes	Yes	Yes	**No**	Yes	Yes
Cefditoren/omeprazole	Yes	Yes	Yes	**No**	**No**	**No**	**No**
Rifampicin/Apixaban	Yes	Yes	Yes	Yes	Yes	Yes	Yes
Fluconazole/Diltiazem	Yes	Yes	**No**	Yes	Yes	Yes	Yes
**Sensitivity**	100%	100%	80%	60%	60%	70%	40%

**Table 4 antibiotics-09-00019-t004:** Overall and partial performance scores of drug–drug interaction software platforms for the detection and guidance on antimicrobial drug interactions.

DDI Software Platform	Sensitivity (%)	Dimensions						Overall Score
		Sensitivity Score *	R.A.	Clinical Effect	M.I.	Evidence	References	
**Lexicomp Drug Interactions**	**100**	**3**	**3**	**2**	**0.5**	**0.5**	**1**	**10**
**Drugs.com**	100	3	2.7	1.6	0.5	0	1	**8.8**
**Epocrates**	80	2	2	1	0.3	0	0	**5.3**
**Medscape**	60	1	2.5	1.1	0.2	0	0	**4.8**
**iDoctus**	60	1	2	1.6	0.5	0.5	1	**6.6**
**Medimecum**	70	1	2.3	1	0.1	0	0	**4.4**
**Guia IF**	40	0	2.2	1.5	0.3	0.1	1	**5.1**

* Sensitivity was scored as three if the DDI checker detected all ADIs, as two if at least eight ADIs were detected, as one if at least six ADIs were detected and as 0 if less than six ADIs were detected. For each of the remaining dimensions, the mean score was calculated considering the partial score of every detected ADI. ** R.A: Recommended actions. It was scored up to three points over the ADIs detected (the same recommendation than the gold standard:1 point; other approximate recommendation: 0.5 points; no recommendation: 0 points). *** Clinical effect: type of toxicity and severity. It was scored up to two points (the same type of toxicity and severity as the gold standard: 1 point (0.5 + 0.5); another type of toxicity or severity: 0 points; no clinical effect: 0 points). **** M.I: mechanism of interaction, scored up to 0.5 points if it is generated by the DDI software platform. ***** Quality of evidence scored up to 0.5 points if it is available. ****** Availability of references scored up to one point.

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
