# Peer review of "Performance Assessment of Software to Detect and Assist Prescribers with Antimicrobial Drug Interactions: Are all of them Created Equal?"

_antibiotics, 2020, doi:10.3390/antibiotics9010019_

Round 1

Reviewer 1 Report

Morte-Romea et. al. review online and standalone software to check antimicrobial drug-drug interactions. Authors conclude drugs.com a free web based software portal is best in providing A DDI. Although I find article useful, the article is little confusing in its methodology.

Authors need to address the following:

How many drugs were queried on each of the software. Why sensitivity of the drugs.com is 100% , is it because rest are normalized treating drugs.com as maximum or was it each DDI was appropriately scored in drugs.com. This needs to be explained!

Reviewer 2 Report

Performance Assessment of Software to Detect and Assist Prescribers with Antimicrobial Drug Interactions: Are all of them Created Equal?

Major Comments:

The subject of this article is consistent with the scope of Antibiotics. After a brief literature search, it seems that there is a paucity of data that investigates differences in antimicrobial drug interaction software, and this is a topic that is clinically relevant. However, this article requires major edits to be fit for publication.

Minor Comments:

Introduction

Line 48: Change "…and cause hospital admissions and visits to emergency room" to "…and increase hospital admissions and emergency room visits". Line 50: Change "Antimicrobials can be involved in multiple DDI" to "…multiple DDIs" Line 52: Change "… favoring drug toxic effects" to "…favoring toxic side effects". Line 54: Change "…and the need to use second line drugs, often less efficacious and more toxic" to "and the need to use sometimes less efficacious and more toxic second line drugs". Line 60: Change "…this kind of tools" to "these tools" Line 63: Change "researched that up to 33%" to "found that up to 33%"

Methods

Move entire methods section to above results section. Line 160: Fix spacing before "Medimecum". Line 162: Make Lexicomp all one word. Table 4: Fix spacing overall in table, delete ":" after trimethoprim. Table 4: Fix typo in "unknown". Line 176: Define what you meant by "accuracy and completeness" and "user experience". Line 181-183: Define how the investigators determined their score of 0 to 5? Was a scale used? Were the definitions of each score consistent? Line 189: Change "others characteristics" to "other characteristics". Data analysis: it seems as though these scales are completely arbitrary. Clarify why 2 points? Why 3 points?

Results

Table 1: It is mentioned that Lexicomp was deemed to be the gold standard, yet this platform is not compared to the others? I would recommend comparing these other platforms to Lexicomp. Same comment for Table 2 also. Table 1: alter alignment and spacing to be consistent across entire table – i.e. align top middle or bottom for all rows and fix spacing for "Recommendations for clinical management" and "Display of the level of evidence". Table 1: What does online vs. offline mean? Does this mean a desktop computer vs. a smartphone app? Please clarify in table. Table 1: For date of last update, there is a * - what does this indicate? Line 75: Fix spacing for "50%" Table 2: fix "Medscape" heading to be all on one line Table 2: What does the heading "Program" mean? These seem to be drug-drug interactions not programs Table 2: Under "quinolones/non-steroidal" – clarify is this supposed to be NSAIDs? Line 77: Change "show" to "shown" Line 78: grammatically reword "No DDI software platform reached the maximum possible 77 score in the “clinical effect” and “recommended action/s” dimensions, being Drugs.com the one that 78 reached highest in both categories." Table 3: Change all words in the headings to be on one line Table 3: How were these scores calculated? For example what made a platform score 0.14 on mechanism vs. 0.5? Figure 1: What does usability and clarity mean? How were these defined? What is this scale going up to 5?

Discussion

Line 114: Change "this type of interactions" to "these types of interactions" Line 118: Again, the "gold standard" is mentioned but the reader is not informed how the gold standard ranks in each of these categories. Examples may be helpful to see how the authors scored each of these platforms. Line 141: Change "mobile apps" to "mobile applications" Line 141-142: meaning here is unclear – please re-word or clarify.

Round 2

Reviewer 2 Report

Overall the authors have sufficiently altered this paper to make it fit for publication - the minor edits below should be addressed prior to publication.

-Line 118: change partial punctuation to partial score

-Supplement material in Lexicomp column - typo in "absorption"

-Throughout the tables and in the supplement: some decimals are reported with commas, please change to periods. ex) 0,5 to 0.5

-Line 157: Change "the same type of toxicity and severity than the gold standard" to "...as the gold standard"